# Alchemy: A benchmark and analysis toolkit for meta-reinforcement learning agents

**Jane X. Wang**[*†1] , **Michael King**[*1], **Nicolas Porcel**[1], **Zeb Kurth-Nelson**[1,2],
**Tina Zhu**[1], **Charlie Deck**[1], **Peter Choy**[1], **Mary Cassin**[1], **Malcolm Reynolds**[1],
**Francis Song**[1], **Gavin Buttimore**[1], **David P. Reichert**[1], **Neil Rabinowitz**[1],
**Loic Matthey**[1], **Demis Hassabis**[1], **Alexander Lerchner**[1], **Matthew Botvinick**[‡1,2]

[1]DeepMind, London, UK
[2]University College London, London, UK

## Abstract

There has been rapidly growing interest in meta-learning as a method for increasing the flexibility and sample efficiency of reinforcement learning. One problem in this area of research, however, has been a scarcity of adequate benchmark tasks. In general, the structure underlying past benchmarks has either been too simple to be inherently interesting, or too ill-defined to support principled analysis. In the present work, we introduce a new benchmark for meta-RL research, emphasizing transparency and potential for in-depth analysis as well as structural richness. Alchemy is a 3D video game, implemented in Unity, which involves a latent causal structure that is resampled procedurally from episode to episode, affording structure learning, online inference, hypothesis testing and action sequencing based on abstract domain knowledge. We evaluate a pair of powerful RL agents on Alchemy and present an in-depth analysis of one of these agents. Results clearly indicate a frank and specific failure of meta-learning, providing validation for Alchemy as a challenging benchmark for meta-RL. Concurrent with this report, we are releasing Alchemy as public resource, together with a suite of analysis tools and sample agent trajectories.

## 1 Introduction

Techniques for deep reinforcement learning have matured rapidly over the last few years, yielding high levels of performance in tasks ranging from chess and Go [53] to realtime strategy [66, 42] to first person 3D games [28, 69]. However, despite these successes, poor sample efficiency, generalization, and transfer remain widely acknowledged pitfalls. To address those challenges, there has recently been growing interest in the topic of meta-learning [7, 64], and how meta-learning abilities can be integrated into deep RL agents [67, 6]. Although a bevy of interesting and innovative techniques for meta-reinforcement learning have been proposed [e.g., 19, 71, 47, 57], research in this area has been hindered by a 'problem problem,' that is, a dearth of ideal task benchmarks. In the present work, we contribute toward a remedy, by introducing and publicly releasing a new and principled benchmark for meta-RL research.

Where deep RL requires a task, meta-RL instead requires a *task distribution*, a large set of tasks with some form of shared structure. Meta-RL is then defined as any process that yields faster learning, on average, with each new draw from the task distribution [60, 51, 5]. A straightforward way to

---

[*]Equal contribution
[†]Correspondence to: wangjane@google.com
[‡]Correspondence to: botvinick@google.com

35th Conference on Neural Information Processing Systems (NeurIPS 2021) Track on Datasets and Benchmarks.

generalize the problem setting is in terms of an underspecified partially observable Markov decision problem [UPOMDP; 13]. This enriches the standard POMDP tuple $\langle S, A, \Omega, T, R, O \rangle$, respectively a set of states, actions, and observations, together with state-transition, reward and observation functions [58], adding a set of parameters $\Theta$ which govern the latter three functions. Importantly, $\Theta$ is understood as a random variable, governed by a prior distribution and resulting in a corresponding distribution of POMDPs. In this setting, meta-RL can be viewed in terms of hierarchical Bayesian inference, with a relatively slow process, spanning samples, gradually inferring the structure of the parameterization $\Theta$, and in turn supporting a rapid process which infers the specific parameters underlying new draws from the task distribution [43, 23, 15, 1]. In this way, meta-RL is equivalent to latent structure learning, as it is the structure in the tasks that allows for fast adaptation in the inner loop. In black-box approaches to meta-RL, this fast process is active, involving strategic information gathering or experimentation [18, 11]. In the limit of training, such approaches are able to implement Bayes-optimal policies due to their meta-learning objective [39, 43], and analysis of their internal representations shows that they come to represent the task in a similar way to Bayes-optimal agents, capturing the necessary and latent structure of the task family.

This perspective brings into view two further desiderata for any benchmark meta-RL task distribution. First, the ground-truth parameterization of the distribution should ideally be *accessible*. This allows agent performance to be compared directly against an optimal baseline, which is precisely a Bayesian learner, sometimes referred to as an 'ideal observer' [20, 43]. Second, the structure of the task distribution should be *interesting*, in that it displays properties comparable to those involved in many challenging real-world tasks. Intuitively, in the limit, interesting structure should feature compositionality, causal relationships, and opportunities for conceptual abstraction [36], and result in tasks whose diagnosis and solutions require strategic sequencing of actions.

Unfortunately, the environments employed in previous meta-RL research have tended to satisfy one of the above desiderata at the expense of the other (see Section 2 on related work). Here, we introduce a task distribution that checks both boxes, offering both accessibility and interestingness, and thus a best-of-both-worlds benchmark for meta-RL research. *Alchemy* is a 3D, first-person perspective video game implemented in the Unity game engine (www.unity.com). It has a highly structured and non-trivial latent causal structure which is resampled every time the game is played, requiring knowledge-based experimentation and strategic action sequencing. Because levels are procedurally created based on a fully accessible, compositional, generative process with well-defined parameterization, we are able to implement a Bayesian ideal observer as a gold standard for performance.

In addition to introducing the Alchemy environment, we evaluate it on two recently introduced, powerful deep RL agents, demonstrating a striking failure of structure learning. Applying a battery of performance probes and analyses to one agent, we provide evidence that its performance reflects a superficial, structure-blind heuristic strategy. Further experiments show that this outcome is not purely due to the sensorimotor complexities of the task, nor to the demands of multi-step decision making. In sum, the limited meta-learning performance appears to be specifically tied to the identification of latent structure, validating the utility of Alchemy as an assay for this particular ability.

## 2 Related work

Where deep RL requires a task, meta-RL instead requires a *task distribution*, a large set of tasks with some form of shared structure. Meta-RL is then defined as any process that yields faster learning, on average, with each new draw from the task distribution [60, 51, 5]. Gradient-based meta-learning approaches such as MAML [19] and variants [e.g., 38, 24, 19] have proven to be quite popular and versatile, but have trouble balancing targeted exploration and exploitation, especially when latent structure is present. Black-box meta-learners, on the other hand, can optimize exploration with exploitation and capitalize on latent task structure to learn efficient exploration end-to-end [68, 14, 50, 40, 56, 27, 73], usually via an autoregressive model such as LSTMs [26] or self-attention [40, 65]. Others have focused on decoupling the exploration issue entirely [37, 47] by learning to predict the task ID, but it remains to be seen if these approaches work when the task distribution has latent structure that is procedurally generated and compositional, as is the case with Alchemy.

A classic example task requiring balanced exploration-exploitation, showcased in studies implementing black-box meta-RL approaches [e.g., 68, 14, 40], is a distribution of bandit problems, each with its own sampled set of action-contingent reward probabilities. Bandits are useful in that they have furnished accessibility, allowing for principled analysis and interpretation of meta-learning performance

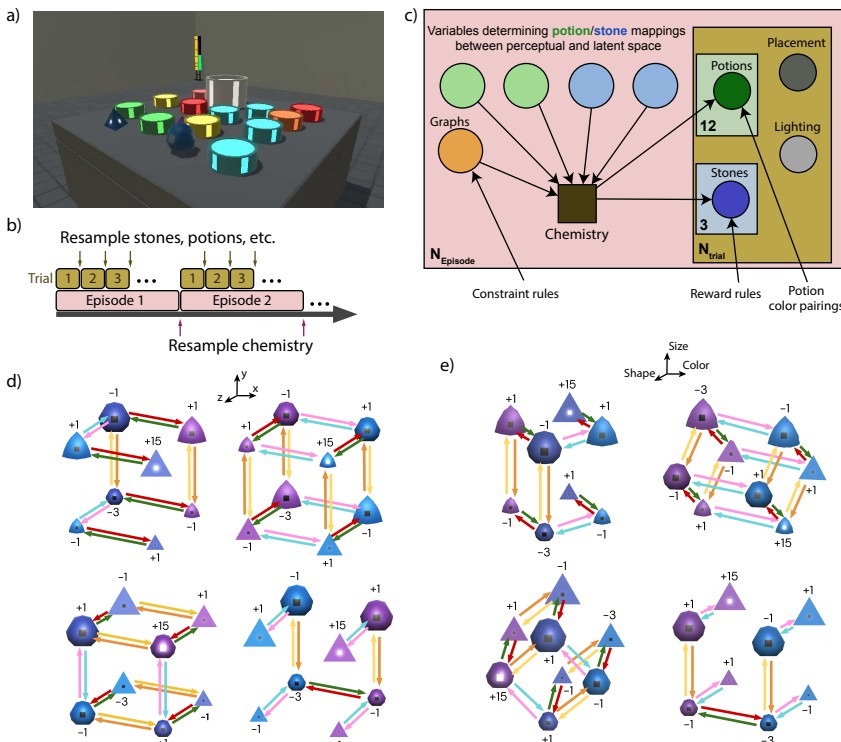

Figure 1: a) Visual observation for Alchemy, RGB, rendered at higher resolution than is received by the agent (96x72). b) Temporal depiction of the generative process, indicating when chemistries and trial-specific instances (stones, potions, placement, lighting) are resampled. c) A high-level depiction of the generative process for sampling a new task, in plate notation. Certain structures are fixed for all episodes, such as the constraint rules governing the possible graphs, the way rewards are calculated from stone latent properties, and the fact that potions are paired into opposites. Every episode, a graph and a set of variables determining the way potions and stones are mapped between latent and perceptual space are sampled to form a new chemistry. Conditioned on this chemistry, for each of $N_{trial} = 10$ trials, $N_s = 3$ specific stones and $N_p = 12$ potions are sampled, as well as random position and lighting conditions, to form the perceptual observation for the agent. See Appendix A.1 and Figure 6 for more details. d) Four example chemistries, in which the latent axes are held constant (worst stone is at the origin). e) The same four chemistries, but with perceptual axes held constant. Note that the edges of the causal graph need not be axis aligned with the perceptual axes.

[68, 14, 43] via comparison with known optimal solutions [see, e.g., 21, 3]. However, they fail on the interestingness front by focusing on very simple task parameterization structures. In Alchemy, we are in a position to compare agent performance against an optimal baseline in the context of a much more interesting – compositional, causal, and conceptual – task distribution. Further, Alchemy was designed with inspiration from cognitive science studies of human behavior [36, 59], where it is argued that latent structure learning is generic to real-world tasks and human-like intelligence. It therefore shares many properties with tasks that evoke human-like cognition, such when children learn to experiment in their environment to learn about causal dependencies [22], or tasks that require theory formation, causal understanding, or planning in a latent space [11, 62]. As with bandits, behavioral tasks in cognitive science also focus on accessibility and understanding, particularly animal tasks, which prioritize interpretability and modeling underlying mechanisms [17, 12] over complexity, and thus are relatively simple [32], especially compared to tasks in AI.

These kinds of properties are notably absent in the most popular meta-RL benchmarks today, which mostly focus on efficient few-shot adaptation of control policies (e.g. MuJoCo continuous control tasks [61, 63] or MetaWorld [72]), in which latent aspects of the task are limited to a single parameter such as goal location. Strategic, smart exploration and latent structure learning are thus not required. At the other end of the spectrum, tasks with more interesting and diverse structure (e.g., Atari [4] or CoinRun [9]) have been grouped together as task distributions, but the underlying structure or

parameterization of those distributions is not transparent [4, 45, 49, 41]. Environments have also been created with emphasis on increasing amounts of complexity and exploration demands, such as NetHack [35], ALFRED [54], Procgen [8], DeepMind Lab [2], interfaces to Minecraft [31], and Obstacle Tower [34], often via procedural generation. Training on procedurally generated tasks improves generalization performance [48, 8, 9] by offering high variety and complexity, but at the cost of these tasks being unable to be transparently analyzed. This makes it difficult to know (beyond human intuition) whether a sample from the task distribution supports transfer to further sampled tasks, let alone to construct Bayes-optimal performance baselines for such transfer. Thus, the current state of meta-RL benchmarks is focused primarily on either efficient adaptation, or enhancing complexity and exploration, at the cost of comprehensibility. Alchemy is distinguished in the compositional and latent structure in which it's procedurally generated, in a manner amenable to analysis—similar to bandits, but more complex—and is focused on latent structure learning. However, we note that we are not the first to emphasize interpretability and diagnosis of certain skills [70, 44, 25], and hope to see more benchmarks emerge with this same philosophy.

## 3 The Alchemy environment

Alchemy is a 3D environment created in Unity [33], played in a series of 'trials', which fit together into 'episodes.' Within each trial, the goal is to use a set of potions to transform each in a collection of visually distinctive stones into more valuable forms, collecting points when the stones are dropped into a central cauldron. The value of each stone is tied to its perceptual features, but this relationship changes from episode to episode, as do the potions' transformative effects. Together, these structural aspects of the task constitute a 'chemistry' that is fixed across trials within an episode, but resampled at the start of each episode, based on a highly structured, compositional, generative process (see Figure 1). The implicit challenge within each episode is thus to diagnose, within the available time, the current chemistry, leveraging this diagnosis to manufacture the most valuable stones possible.

### 3.1 Observations, actions, and task logic

At the beginning of each trial, the agent views a table containing a cauldron together with three stones and twelve potions, as sampled from Alchemy's generative process (Figure 1a). Stone appearance varies along three feature dimensions: size, color and shape. Each stone also displays a marker whose brightness signals its point value (-3, -1, +1 or +15). Each potion appears in one of six hues. The agent receives 96x72 RGB pixel observations from an egocentric perspective, together with proprioceptive information (acceleration, distance of and force on the hand, and whether the hand is grasping an object). It selects actions from a nine-dimensional set (consisting of navigation, object manipulation actuators, and a discrete *grab* action). When a stone comes into contact with a potion, the latter is consumed and, depending on the current chemistry, the stone appearance and value may change. Each trial lasts sixty seconds, simulated at 30 frames per second. A visual indicator in each corner of the workspace indicates time remaining. Each episode comprises ten trials, with the chemistry fixed across trials but stone and potion instances, spatial positions, and lighting resampled at the onset of each trial (Figure 1b). See `https://youtu.be/k2ziWeyMxAk` for a game play video.

### 3.2 The chemistry

As noted earlier, the causal structure of the task changes across episodes. The current 'chemistry' determines the particular stone appearances that can occur, the value attached to each appearance, and, crucially, the transformative effects that potions have on stones. The specific chemistry for each episode is sampled from a structured generative process, illustrated in Figure 1c-e and fully described in the Appendix. For brevity, we limit ourselves here to a high-level description.

To foreground the meta-learning challenge involved in Alchemy, it is useful to distinguish between (1) the aspects of the task that can change across episodes and (2) the abstract principles or regularities that span all episodes. As we have noted, the former, changeable aspects include stone appearances, stone values, and potion effects. Given all possible combinations of these factors, there exist a total of 167,424 possible chemistries (taking into account that stone and potion instances are also sampled per trial yields a total set of possible trial initializations on the order of 124 billion, still neglecting variability in the spatial positioning of objects, lighting, etc.).[4] The principles that span episodes are, in a sense, more important, since identifying and exploiting these regularities is the essence of the

---

[4]Note that this corresponds to the sample space for the parameter set $\Theta$ mentioned in the Introduction.

meta-learning problem. The invariances that characterize the generative process in Alchemy can be summarized as follows:

1. Within each episode, stones with the same visual features have the same value and respond identically to potions. Analogously, potions of the same color have the same effects.

2. Within each episode, only eight stone appearances can occur, and these correspond to the vertices of a cube in the three-dimensional appearance space. Potion effects run only along the edges of this cube, effectively making it a causal graph.

3. Each potion type (color) 'moves' stone appearances in only one direction in appearance space. That is, each potion operates only along parallel edges of the cubic causal graph.

4. Potions come in fixed pairs (red/green, yellow/orange, pink/turquoise) which always have opposite effects. The effect of the red potion, for example, varies across episodes, but whatever its effects, the green potion will have the converse effects.

5. In some chemistries, edges of the causal graph may be missing, i.e., no single potion will effect a transition between two particular stone appearances.[5] However, the topology of the underlying causal graph is not arbitrary; it is governed by a generative grammar that yields a highly structured distribution of topologies (see Appendix A.1 and Figure 7).

These conceptual aspects of the task remain invariant across episodes, so experience gathered from a large set of episodes affords the opportunity for an agent to discover them, tuning into the structure of the generative process giving rise to each episode and trial. It is the ability to learn at this level, and to exploit what it learns, that corresponds to an agent's meta-learning performance.

### 3.3 Symbolic version

As a complement to the canonical 3D version of Alchemy, we have also created a symbolic version of the task. This involves the same underlying generative process and preserves the challenge of reasoning and planning over the resulting latent structure, but factors out the visuospatial and motor complexities of the full environment. Symbolic Alchemy returns as observation a concatenated vector indicating the features of all sampled potions and stones, and entails a discrete action space, specifying a stone and a container (either potion or cauldron) in which to place it, plus a *no-op* action (navigation is not required). Full details are presented in the Appendix.

### 3.4 Ideal-observer reference agent

| Agent | Episode score |
|---|---|
| IMPALA | $140.2 \pm 1.5$ |
| VMPO | $156.2 \pm 1.6$ |
| VMPO (Symbolic) | $155.4 \pm 1.6$ |
| Ideal observer | $284.4 \pm 1.6$ |
| Oracle | $288.5 \pm 1.5$ |
| Random heuristic | $145.7 \pm 1.5$ |

Table 1: Benchmark and baseline-agent evaluation scores (mean $\pm$ standard error over 1000 episodes).

As noted in the Introduction, when a task distribution is fully accessible, this makes it possible to construct a Bayes-optimal 'ideal observer' reference agent as a gold standard for evaluating the meta-learning performance of any agent. We constructed just such an agent for Alchemy, as detailed in the Appendix (Algorithm 1). This agent maintains a belief state over all possible chemistries given the history of observations, and performs an exhaustive search over both available actions (as discretized in symbolic Alchemy) and possible outcomes in order to maximize reward at the end of the current trial. The resulting policy both marks out the highest attainable task score (in expectation) and exposes minimum-regret action sequences, which optimally balance exploration or experimentation against exploitation.[6] Any agent matching the score of the ideal observer

---

[5]When this is the case, in order to anticipate the effect of some potions the player must attend to conjunctions of perceptual features. Missing edges can also create bottlenecks in the causal graph, which make it necessary to first transform a stone to look *less* similar to a goal appearance before it is feasible to attain that goal state.

[6]Our ideal observer does not account for optimal inference over the entire length of the episode, which would be computationally intractable to calculate. However, in general, we find that a single trial is enough to narrow down the number of possible world states to a much smaller number, and thus searching for more than one trial does not confer significant benefits.

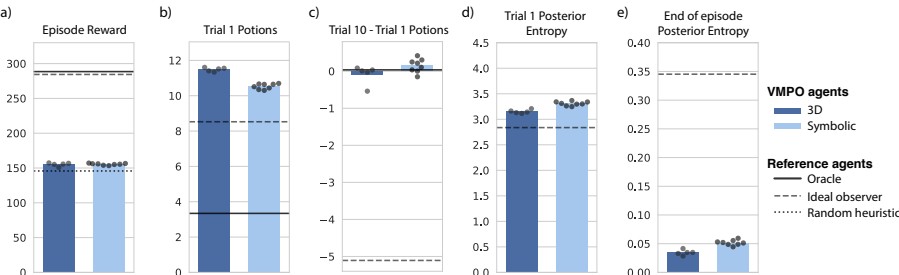

Figure 2: Detailed analysis of behavior of VMPO agents on 1000 evaluation episodes in 3D Alchemy and symbolic Alchemy, in comparison with the reference agents which have been handcrafted with specific knowledge (oracle, Bayesian ideal observer, and random heuristic). a) Episode reward. b) Number of potions used in Trial 1. c) Difference between number of potions used in trial 10 vs trial 1. d) Posterior entropy over world states, conditioned on agent actions, at end of trial 1. e) Posterior entropy over world states at the end of the episode. Filled black circles indicate individual replicas (5-8 per condition). Note that the oracle and random heuristic reference agents don't maintain a belief state and thus are not indicated in d-e.

demonstrates, in doing so, both a thorough understanding of Alchemy's task structure and strong action-sequencing abilities.

As further tools for analysis, we devised two other reference agents. An *oracle* reference agent is given privileged access to a full description of the current chemistry, and performs a brute-force search over the available actions, seeking to maximizing reward (see Appendix, Algorithm 2). A *random heuristic* reference agent chooses a stone at random, using potions at random until that stone reaches the maximum value of +15 points. It then deposits the stone in the cauldron, and repeats with a new one (Appendix, Algorithm 3). The resulting policy yields a score reflecting what is possible in the absence of any guiding understanding of the latent structure of the Alchemy task.

## 4 Experiments

### 4.1 Agent architectures and training

**VMPO agent:** As described in [55] and [46], this agent centered on a gated transformer XL network. Image-frame observations were passed through a residual-network encoder and fully connected layer, with proprioceptive observations, previous action and reward then concatenated. In the symbolic task, observations were passed directly to the transformer core. Losses were included for policy and value heads, pixel control [30], kickstarting [52, 10] and latent state prediction (see Section 4.3). Where kickstarting was used (see below), the loss was $KL(\pi_{student}\|\pi_{teacher})$, and the weighting was set to 0 after 5e8 steps. Further details are presented in Appendix Table 2.

**IMPALA agent:** This agent is described by [16], and used population-based training as presented by [29]. Pixel observations passed through a residual-network encoder and fully connected layer, and proprioceptive observations were concatenated to the input at every spatial location before each residual-network block. The output was fed into the core of the network, an LSTM.[7] Pixel control and kickstarting losses were used, as in the VMPO agent. See Appendix Table 3 for details.

Both agents were trained for 2e10 steps (4.44e6 training episodes; 1e9 episodes for the symbolic version of the task), and evaluated without weight updates on 1000 test episodes. Note that to surpass a zero score in full, 3D, image-based Alchemy, both agents required kickstarting [52], with agents first trained on a fixed chemistry with shaping rewards included for each potion use. Agents trained on the symbolic version of the task were trained from scratch without kickstarting.

### 4.2 Agent performance and diagnostic analyses

Mean episode scores for both agents are shown in Table 1. Both fell far short of the gold-standard ideal observer benchmark, implying a failure of meta-learning. In fact, scores for both agents fell close to that attained by the random heuristic reference policy. In order to better understand these results,

---

[7]Recurrent state was set to zero at episode boundaries, but not between trials, enabling the agents (in principle) to utilise knowledge of the chemistry accumulated over previous trials.

we conducted a series of additional analyses, focusing on the VMPO agent given its slightly higher baseline score. These analyses are based on a principle of progressively allowing the agent access to different forms of privileged information about the environment in order to pinpoint the source of difficulty. Importantly, note that having access to these kinds of information is not considered 'solving' Alchemy, but rather is merely diagnostic of the specific limitations of current agents, which are falling short in ways we detail below.

A first question was whether this agent's poor performance is due either to the difficulty of discerning task structure through the complexity of high-dimensional pixel observations, or to the challenge of sequencing actions in order to capitalize on inferences concerning the task's latent state. A clear answer is provided by the scores from a VMPO agent trained and tested on the symbolic version of Alchemy, which lifts both of these challenges while continuing to impose the task's more fundamental requirement for structure learning and latent-state inference. As shown in Table 1 and Figure 2a, performance was no better in this setting than in the canonical version of the task, again only slightly surpassing the score from the random heuristic policy.

Informal inspection of trajectories from the symbolic version of the task was consistent with the conclusion that the agent, like the random heuristic policy, was dipping stones in potions essentially at random until a high point value happened to be attained. To test this impression more rigorously, we measured how many potions the agent consumed, on average, during the first and last trials within an episode. As shown in Figure 2b, the agent used more potions during episode-initial trials than the ideal observer benchmark. From Figure 2c, we can see that the ideal observer used a smaller number of potions in the episode-final trial than in the initial trial, while the VMPO baseline agent showed no such reduction (see Appendix Figure 10 for more detailed results). By selecting diagnostic actions, the ideal observer progressively reduces its uncertainty over the current latent state of the task (i.e., the set of chemistries possibly currently in effect, given the history of observations). This is shown in Figure 2d-e, in units of posterior entropy, calculated as the log of the number of states still possible, according to the ideal observer. The VMPO agent's actions also effectively reveal the chemistry, as indicated in the figure. The fact that the agent is nonetheless scoring poorly and overusing potions implies that it is failing to make use of the information its actions have inadvertently revealed about the task's latent state.

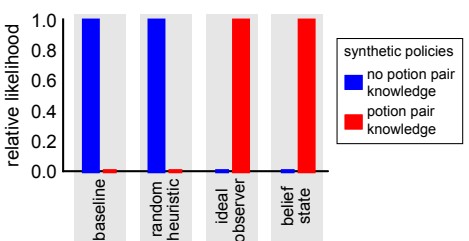

Figure 3: Bayesian model comparison. Two synthetic, probabilistic policies were crafted and compared with the actions of VMPO and reference agents: (1) a policy that does not assume potions come in pairs with opposite effects (blue bars), and (2) a policy that does (red bars). Comparing the goodness of fit between these policies, we found that both the baseline VMPO agent and the random heuristic were better fit by model (1), while the VMPO agent which had belief state extra input was better fit by model (2), and thus appeared to exploit knowledge of potion pairs.

The behavior of the VMPO agent suggests that it has not tuned into the consistent principles that span chemistries in Alchemy, as enumerated in Section 3.2. One way of probing the agent's 'understanding' of individual principles is to test how well its behavior is fit by synthetic, adaptive, probabilistic policies that either do or do not leverage one relevant aspect of the task's structure, a technique frequently used in cognitive science to analyze human learning and decision making [12, 17]. We applied this strategy to evaluate whether agents trained on the symbolic version of Alchemy was leveraging the fact that potions come in consistent pairs with opposite effects (see Section 3.2). Two hand-crafted, probabilistic policies were devised, both of which performed single-step look-ahead to predict the outcome (stones and potions remaining) for each currently available action, attaching to each outcome a value equal to the sum of point-values for stones present, and selecting the subsequent action based on a soft-max over the resulting state values. In both policies, predictions were based on a posterior distribution over the current chemistry. However, in one policy this posterior was updated with built-in knowledge of the potion pairings, while the other policy ignored this regularity of the task. As shown in Figure 3, when the predictions of these two policies were compared to the actions of the ideal observer reference agent, in terms of relative likelihood, results clearly indicated a superior fit for the policy leveraging knowledge of the potion pairings. In contrast, for the random heuristic

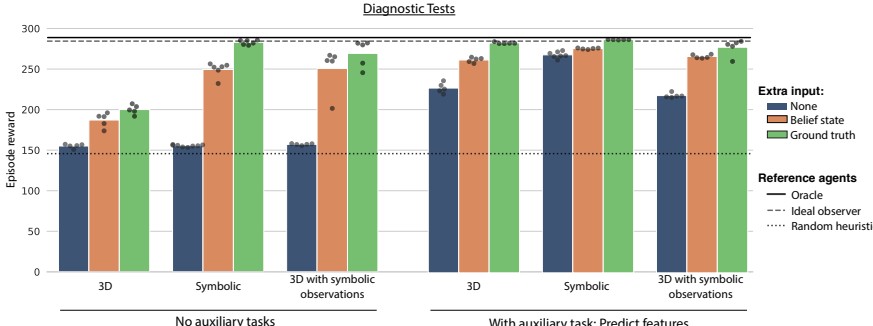

Figure 4: Evaluation performance of VMPO agents under additional diagnostic conditions (not considered part of the Alchemy benchmark). Orange bars add extra input with belief state of ideal observer. Green bars add extra input with ground truth chemistry knowledge. In "3D with symbolic observations" condition, agent acts in 3D world, but is also given the observations it would receive in the corresponding configuration of the symbolic task. Finally, the right group of 9 bars adds an auxiliary feature prediction task. Filled black circles indicate individual replicas (5-8 per condition).

reference agent, a much better fit was attained for the policy operating in ignorance of the pairings. Applying the same analysis to the behavior of the baseline VMPO agent yielded results mirroring those for the random heuristic agent (see Figure 3 'baseline'), consistent with the conclusion that the VMPO agent's policy made no strategic use of the existence of consistent relationships between the potions' effects.

## 4.3 Augmentation studies

A standard strategy in reinforcement learning research is analyzing the operation of a performant agent via a set of ablations, to determine what factors are causal in the agent's success. Confronted with a poorly performing agent, we inverted this strategy, undertaking a set of *augmentations* (additions to either the task or the agent) in order to identify what factors might be holding the agent back.

Given the failure of the VMPO agent to show signs of identifying the latent structure of the Alchemy task, one question of clear interest is whether performance would improve if the task's latent state were rendered observable. In order to study this, we trained and tested on a version of symbolic Alchemy which supplemented the agent's observations with a binary vector indicating the complete current chemistry (see Appendix for details). When furnished with this additional information, the agent's average score jumped dramatically, landing very near the score attained by the ideal observer reference (Figure 4 'Symbolic', 'Extra input: Ground truth'). Note that this augmentation gives the agent privileged access to an oracle-like indication of the current ground-truth chemistry. In a less drastic augmentation, we replaced this additional input with a vector indicating not the ground-truth chemistry, but instead the set of chemistries consistent with observations made so far in the episode, corresponding to the Bayesian belief state of the ideal observer reference model ('Extra input: Belief state'). While the resulting scores in this setting were not quite as high as those in the ground-truth augmentation experiment, they were much higher than those observed without augmentation (i.e. 'Extra input: None'). Furthermore, the agent furnished with the belief state resembled the ideal observer agent in showing a clear reduction in potion use between the first and last trials in an episode (Figure 5a-b).[8] Model fits also indicated the agent receiving this input made use of the opposite-effects pairings of potions, an ability not seen in the baseline VMPO agent (Figure 3).

The impact of augmenting the input with an explicit representation of the chemistry implies that the VMPO agent, while evidently unable to identify Alchemy's latent structure, can act adaptively if that latent structure is helpfully brought to the surface. Since this observation was made in the setting of the symbolic version of Alchemy, we tested the same question in the full 3D version of the task. Interestingly, the results here were somewhat different: While appending a representation of either the ground-truth chemistry or belief state to the agent's observations did increase scores, the effect

---

[8]In contrast, when the agent was furnished with the full ground truth, it neither reduced its potion use over time nor so effectively narrowed down the set of possible chemistries. This makes sense, however, given that full knowledge of the current chemistry allows the agent to be highly efficient in potion use from the start of the episode, and relieves it from having to perform diagnostic actions to uncover the current latent state.

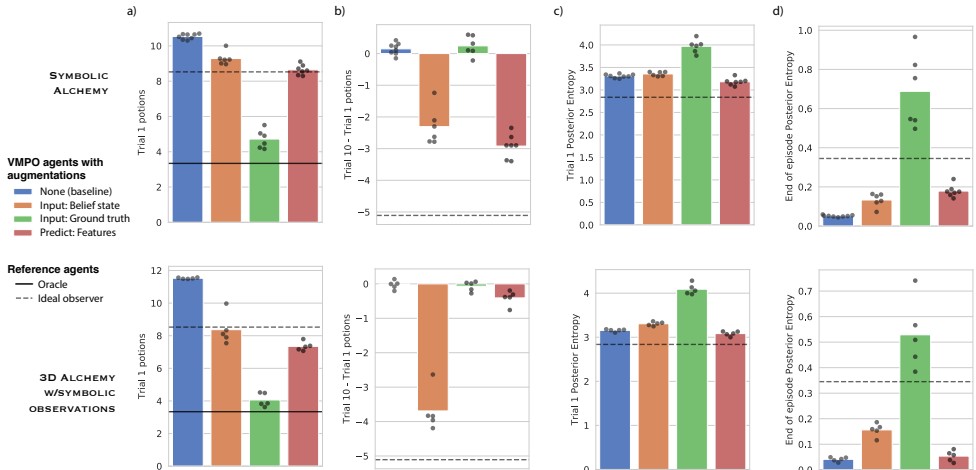

Figure 5: Behavioral metrics for agents trained under different diagnostic conditions and augmentations in symbolic alchemy (top) and 3D alchemy with symbolic observations (bottom). a) Number of potions used in Trial 1. b) Difference between number of potions used in trial 10 vs trial 1. c) Posterior entropy over world states, conditioned on agent actions, at end of trial 1. d) Posterior entropy over world states at the end of the episode. Filled black circles are individual replicas.

was not as categorical as in the symbolic setting (Figure 4 '3D'). Two hypotheses suggest themselves as explanations for this result. First, the VMPO agent might have more trouble capitalizing on the extra information in the 3D version of Alchemy because doing so requires composing much longer sequences of action than does the symbolic version of the task (due to the need to navigate around in a 3D environment). Second, the greater complexity of the agent's perceptual observations might make it harder to map this extra information onto the structure of its current perceptual inputs. As one step toward adjudicating between these possibilities, we augmented the inputs to the agent in the 3D task with the observations that the symbolic version of the task would provide in the same state. In the absence of extra input about the currently prevailing chemistry, this augmentation did not change the agent's behavior; the resulting score still fell close to the random reference policy (Figure 4 '3D with symbolic observations'). However, adding either the ground-truth or belief-state input raised scores much more dramatically than when those inputs were included without symbolic state information, elevating them to levels comparable to those attained in the symbolic task itself and approaching the ideal observer model, with parallel effects on potion use (Figure 5a-b). These results suggest that the failure of the agent to fully utilize this extra information about the current chemistry was not due to challenges of action sequencing in the full 3D version of Alchemy, but stemmed instead from an inability to effectively map such information onto internal representations of the current perceptual observation.

While augmenting the agent's inputs is one way to impact its representation of current visual inputs, the recent deep RL literature suggests a different method for enriching internal representations, which is to add auxiliary tasks to the agent's training objective [see, e.g., 30]. As one application of this idea, we added to the RL objective a set of supervised tasks, the objectives of which were to produce outputs indicating (1) the total number of stones present in each of a set of perceptual categories, (2) the total number of potions present of each color, and (3) the currently prevailing ground-truth chemistry (see Appendix B.2 for further details).[9]

Introducing these auxiliary tasks had a dramatic impact on agent performance, especially for (1) and (2) (less so for (3), see Figure 9 in Appendix). This was true even in the absence of extra belief-state or ground-truth input, but the most dramatic benefits were achieved when both forms of augmentation were present, yielding the highest scores observed so far in the full 3D version of Alchemy (Figure 4 '3D'). Indeed, in the presence of the auxiliary tasks, further supplementing the inputs with the symbolic version of the perceptual observations added little to agent performance (Figure 4 '3D with symbolic observations'). Adding the auxiliary tasks to the objective in symbolic

---

[9]Prediction tasks (1) and (2) were always done in conjunction and are collectively referred to as 'Predict: Features', while (3) is referred to as 'Predict: Chemistry'.

Alchemy had a striking effect on scores even in the absence of any other augmentation. In this case, scores approached the ideal observer benchmark (Figure 4 'Symbolic'), providing the only case in the present study where agents showed respectable meta-learning performance on either version of Alchemy without privileged information at test.[10]

# 5   Discussion

We have introduced Alchemy, a new benchmark task environment for meta-RL research. Alchemy is novel among existing benchmarks in bringing together two desirable features: (1) structural *interestingness*, due to its abstract, causal, and compositional latent organization, which demands experimentation, structured inference and strategic action sequencing; and (2) structural *accessibility*, conferred by its explicitly defined generative process, which furnishes an interpretable prior and supports construction of a Bayes-optimal reference policy, alongside many other analytical maneuvers. With the hope that Alchemy will be useful to the larger community, we are releasing, open-source, both the full 3D and symbolic versions of the Alchemy environment, along with a suite of benchmark policies, analysis tools, and episode logs (https://github.com/deepmind/dm_alchemy).

As a first application and validation of Alchemy, we tested two strong deep RL agents which are considered highly performant [16, 46, 55]. In both cases, despite mastering the basic mechanical aspects of the task, neither agent showed any appreciable signs of meta-learning (i.e. structure learning or latent-state inference), which we ascertained through a series of analyses made possible by the task's accessible structure and comparison with reference agents. Leveraging a symbolic version of Alchemy, we were able to establish that this failure of meta-learning is not due purely to the visual complexity of the task or to the number of actions required to achieve task goals. Finally, a series of augmentation studies showed that deep RL agents can in fact perform well if the latent structure of the task is rendered fully observable, especially if auxiliary tasks are introduced to support representation learning. These insights may, we hope, be useful in developing deep RL agents that are capable of solving Alchemy without access to privileged information. Due to the depth of analysis, this work was limited in only thoroughly characterizing the behavior of one deep RL agent (VMPO). Future work is needed to apply the same level of careful analysis across a broader range of deep RL agents, although we're hopeful that with the release of this environment and analysis tools, other researchers will find it relatively easy to train and analyze their own agents.

Alchemy is, admittedly, specific in its construction, and therefore limited in terms of its complexity, relative to real-world or other tasks. This was necessary in order to preserve its accessibility, but reduces its ability to challenge agents along other dimensions (such as extended exploration). Another limitation is that extended training can be required, especially for the 3D version. It was necessary to use kickstarting from simpler, reward-shaped versions of the task in order to gain enough signal to overcome the visuomotor difficulties. Due to Alchemy's partial observability, even the symbolic version can be quite challenging (although it is possible to significantly simplify the environment by holding fixed various aspects of the generative chemistry).

We note that Alchemy is not simply a benchmark to evaluate performance, but a comprehensive analysis suite to help characterize behavior and pinpoint points of failure. We demonstrate a range of detailed, hypothesis-driven, and informative analyses enabled by this testbed, focusing on depth rather than breadth, in order to showcase the extent of what is possible. This is a distinctly cognitive science style of analysis, and is in contrast with performance-focused benchmarks which offer limited potential for analysis or insight into agent behavior (although we are encouraged to see the emergence of new cognitive science-inspired works such as [70]). Importantly, in our view, the main contributions of the present work inhere not in the specific concrete details of Alchemy itself, but rather in the overall scientific agenda and approach. Ascertaining the level of knowledge possessed by deep RL agents is a challenging task, comparable to trying to ascertain the knowledge of real animals, and (as in that latter case) requiring detailed cognitive modeling. Alchemy is designed to make this kind of modeling not only possible, but even central, and we propose that more meta-RL environments should strive to afford the same granularity of insight into agent behavior. As a closing remark, we have found that many humans find playing Alchemy interesting and challenging, and our informal tests suggest that motivated players, with sufficient training, can attain to sophisticated strategies and high levels of performance. Based on this, we suggest that Alchemy may also be an interesting task for research into human learning and decision making.

---

[10]Potion use, as well as world state uncertainty, also showed a reduction over trials (see Figure 5).

## Acknowledgments and Disclosure of Funding

We would like to thank the DeepMind Worlds team – in particular, Ricardo Barreira, Kieran Connell, Tom Ward, Manuel Sanchez, Mimi Jasarevic, and Jason Sanmiya for help with releasing the environment and testing, and Sarah York for her help on testing and gathering human benchmarks. We also are grateful to Sam Gershman, Murray Shanahan, Irina Higgins, Christopher Summerfield, Jessica Hamrick, David Raposo, Laurent Sartran, Razvan Pascanu, Alex Cullum, and Victoria Langston for valuable advisement, discussions, feedback, and support. This work was funded by DeepMind.

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
