# OpenReview forum: "Alchemy: A benchmark and analysis toolkit for meta-reinforcement learning agents"
_NeurIPS.cc/2021/Track/Datasets_and_Benchmarks/Round2 — NeurIPS 2021 Datasets and Benchmarks Track (Round 2)_

### Official Review · Reviewer_WeQ5 · 2021-09-11
**An interesting benchmark, potentially NeurIPS worthy, but clarifications are needed**

**Rating:** 7
**Confidence:** 3
**Clarity:** Excellent writing quality.

**Strengths:**

First of all, note that I am not an expert on Meta-RL, my domain of expertise is more on Automatic Curriculum Learning (ACL) & Deep RL.

This benchmark seems relevant to the meta-RL community, as it proposes a standardized testbed on which to test meta-RL agents. It is well documented and as such it seems accessible for a large audience. Computation costs seems quite high, but a symbolic version of Alchemy is provided, which is good for research teams with low compute budgets.

Actually, the meta-RL objective studied here is quite different from classical Meta-RL works, e.g. MAML, or the Meta-World paper, which focuses on the problem of meta-learning a new task in a few gradient updates. Here agents must learn over the course of an episode, without updates, which is an interesting setup.

Also, since one of Alchemy strengths is to have an accessible parametric task distribution, i.e. a task space, I'm wondering whether this platform could also be used as an ACL benchmark. It would only require the additional assumption that the ACL algorithm is allowed to control the chemistry sampling procedure. Anyway, this is most likely out of the scope of this paper.

**Weaknesses:**

For me a *potential* major weakness is the lack of Meta-RL baselines. VMPO and IMPALA are SOTA Deep RL agents, without meta-RL mechanisms. I am wondering whether there are better meta-RL baselines in the literature ? Maybe not, because the benchmark assumes no gradient updates at test time. If it is the case that there is no alternative meta-RL baseline agent, then I think that the paper would benefit from clearly stating this.

I am looking forward to discuss with authors and other reviewers on this subject. If this problem is clarified, I will gladly increase my score.

**Additional Feedback:**

What is the Bayes optimal agent exactly accessing to ?

**Correctness:**

Evaluation methods are appropriate, clearly stated, and performed correctly. Running more seeds per tested conditions would have been better, as it would have allowed to perform statistical significance tests, but the current status of the manuscript is acceptable w.r.t. the current state of the DRL literature.

**Documentation:**

Documentation is sufficient. I really enjoyed the video. Very informative.

**Relation To Prior Work:**

Related work appears to be properly discussed

**Summary And Contributions:**

Authors propose the Alchemy benchmark, which is an environment with parametric procedural generation, allowing to propose a diversity of complex tasks to meta-RL agents. Using Alchemy, authors conduct experiments with two SOTA Deep RL algorithms, and provide an in-depth analysis of their (low) performance results, showcasing that the meta-RL challenge of Alchemy is relevant.

---

> ### Author Response · Authors · 2021-09-24
> **Response from authors**
>
> We thank the reviewer for their kind comments about our paper, and for the time and effort spent providing useful feedback. We will address the review point-by-point below.
>
> > Actually, the meta-RL objective studied here is quite different from classical Meta-RL works, e.g. MAML, or the Meta-World paper, which focuses on the problem of meta-learning a new task in a few gradient updates. Here agents must learn over the course of an episode, without updates, which is an interesting setup.
>
> As noted in our response to Reviewer 1Tpo, in one of the original definitions of learning to learn, put forward by Thrun and Pratt in 2012 (Learning to Learn, Springer Science & Business Media), all that is required for an algorithm to meta-learn is to train on a family of interrelated tasks and to display improved learning with increased experience and more samples from the task distribution. The gradient-based approach, introduced by Finn et al (2017) in MAML, is actually relatively new, although it has proven to be immensely popular and versatile. The method of meta-learning we focus on is that of black-box, memory-based meta-learning, in which adaptation occurs via a recurrent, internal state, rather than a few gradient updates. Popular examples include Wang et al 2016 (arXiv:1611.05763), Duan et al 2016 (arXiv:1611.02779), Mishra et al 2017 (ICLR), and Santoro et al 2016 (ICML).
>
> > Also, since one of Alchemy strengths is to have an accessible parametric task distribution, i.e. a task space, I'm wondering whether this platform could also be used as an ACL benchmark. It would only require the additional assumption that the ACL algorithm is allowed to control the chemistry sampling procedure. Anyway, this is most likely out of the scope of this paper.
>
> Thank you for the very interesting suggestion! We hadn’t considered this before, but we’re certainly open to future collaborations within this realm.
>
>
> > For me a potential major weakness is the lack of Meta-RL baselines. VMPO and IMPALA are SOTA Deep RL agents, without meta-RL mechanisms. I am wondering whether there are better meta-RL baselines in the literature ? Maybe not, because the benchmark assumes no gradient updates at test time. If it is the case that there is no alternative meta-RL baseline agent, then I think that the paper would benefit from clearly stating this.
>
> We respectfully disagree with the statement that the two agents we tested don’t have meta-RL mechanisms. As noted above, gradient-based meta-learning is only one approach, but meta-learning approaches also include those that are recurrent memory-based or metric based (see Vanschoren 2018 arXiv:1810.03548 for a comprehensive review). Indeed, any architecture with nested optimization loops trained via "meta-episodes" (i.e. in an episodic fashion) on a family of tasks can be considered to have some meta-learning capacity. VMPO and Impala are both implemented with recurrent cores (either transformers or LSTMs), and so both indeed would qualify as being meta-RL agents. As this seems to be a common misconception, we’ll add clarification to the text, and thank you for bringing this point of confusion to our attention.
>
> We hope that this assuages your primary concern, and we would be happy to follow up with any additional concerns or questions.
>
>
> > Evaluation methods are appropriate, clearly stated, and performed correctly. Running more seeds per tested conditions would have been better, as it would have allowed to perform statistical significance tests, but the current status of the manuscript is acceptable w.r.t. the current state of the DRL literature.
>
> We also find that 5-8 seeds (replicas) per condition seems to be at or above the norm for our field. We can certainly do statistical tests and add them to our text. Are there any in particular that you feel would be interesting to see (ie between which conditions)?
>
> > What is the Bayes optimal agent exactly accessing to ?
>
> Is the question regarding what information the Bayes optimal agent has access to? This agent knows about the invariances that are discussed in section 2.2 (e.g. the fact that the potion colors are paired), but needs to explore and gain information in order to know which potions have which effect in any particular episode (that is, it has to figure out what specific chemistry has been sampled). The oracle agent has access to the actual chemistry that has been sampled, and so doesn’t need to do any exploration.

---

> > ### Comment · Reviewer_WeQ5 · 2021-09-27
> > **Thank you for answering**
> >
> > Authors addressed all my concerns.
> >
> > It is now clear (to me) that their considered baselines are relevant.
> >
> > Thank you for your answers.
> >
> > I will raise my score from 5 to 6.

---

> > > ### Author Response · Authors · 2021-09-28
> > > **Revision uploaded**
> > >
> > > Thank you for your kind reconsideration and response.
> > >
> > > We have now uploaded a revised version which has, among other changes, expanded considerably on meta-RL background and related work, placing Alchemy within a much broader context, and hopefully addresses your concerns thoroughly.

---

> > > > ### Comment · Reviewer_WeQ5 · 2021-09-29
> > > > **Paper update**
> > > >
> > > > I confirm that the updated version better situate this benchmark within related meta-RL works.
> > > > All my main concerns have been addressed.
> > > >
> > > > I also note (after re-checking the code release, without running it) that authors made a lot of efforts on the open-source release of their work. The code is clean and commented, they included a google colab tutorial, a very useful environment presentation video, and clear installation instructions.
> > > >
> > > > Considering this, I will further increase my score from 6 to 7, as it is indeed a good benchmark paper.

---

### Official Review · Reviewer_WAyX · 2021-09-19
**Very interesting work that helps perform in-depth analyses for meta-RL; should improve clarity and describe limitations in more detail**

**Rating:** 7
**Confidence:** 4
**Correctness:** I believe the claims are correct.

**Strengths:**

The experiments are very interesting and thorough. The authors study VMPO and IMPALA as baselines and compare their performances with a Bayes ideal observer with a belief state, an oracle and a random heuristic agent. They also augment two versions (image-based and feature-based) of the VMPO baseline with inputs of the belief state/ground truth to analyse the impact of these and show that their performances improve.

They also try out experiments to disentangle whether task sequencing or complexity of visual observations hold back the image-based VMPO agent from utilising privileged information which the feature-based VMPO was able to utilise.

They further try out helping the image-based and feature-based VMPO agents by adding auxilliary tasks of predicting certain features which drastically improve their performances.


**Weaknesses:**

1) **The clarity of the paper could be improved in many places**

I will now use quotes from the text and describe my issues here.
>As shown in Figure 3a, the agent used more potions during episode-initial trials than the ideal observer benchmark.

It is not clear which bar "agent" in the above sentence refers to in Fig. 3a and thus it is difficult to parse the figure. Why not already call the agent by the name used in the legend of the plot? For example, "Input: Ground truth" and "Input: Belief state" haven't been defined in the text yet at this point and I found it hard to make out if "Input: Ground truth" may be the oracle or the ideal observer. This is especially so because the oracle or the ideal observer are not mentioned in the figure caption until the end and I had read the caption only till where it describes sub-figure "a" because the text refers to Fig. 3a. I would recommend to add the ideal observer and oracle also to the graph legend instead of just in the caption.

I think *I* know why the oracle is only shown in sub-figure a in Fig. 3 but it would be nice to add the reasoning behind this to the caption because it took me a while to figure it out.

I did not understand the "Synthetic models" paragraph. Especially, what "fit of these two models" and "was compared for the behavior of the ideal observer reference agent". Additionally, the sentence
>In both models, predictions were based on a posterior distribution over the current chemistry.

could just say that the input was the belief state. Then the terminology would be consistent with the figure caption. in my opinion, scientific papers should try to use consistent terminology to reduce the cognitive load of understanding them.

>When furnished with this side information, the agent’s average score jumped dramatically, landing very near the score attained by the ideal observer reference (Figure 2a ‘Symbolic’).

Did you mean Fig. 2b? I don't see this in 2a (this error is repeated multiple times later). Again, it could have been mentioned here that the reader should look at the "ground truth" bar in the left half of 2b instead of saying "side information". The "ground truth" aspect is only mentioned in the next sentence but I was already trying to parse the figure after reading the quoted sentence above.

>the VMPO agent might have more trouble capitalizing on the given side information in the 3D version of Alchemy because doing so requires composing much longer sequences of action than does the symbolic version of the task.

Why does it require longer sequences?

>compositional latent organization

Compositionality was not discussed in the text. I would like that it is described why Alchemy tasks feature compositionality.

I also find the term "3D Alchemy" misleading because I think the authors mean the image-based task here but according to me even the symbolic version is inherently describing a 3D world and thus I find "3D Alchemy" a misnomer.

What does "individual replicas" mean? Seeds? Why are they 5-8 and not a fixed number?

In line 300, please mention what part of the Appendix.

Sentence beginning in lines 300-306 could be improved grammatically.

>a technique frequently used in cognitive science to analyze human learning and decision making

Citation needed.

Figures 3 and 4 are discussed before Figure 2. And it is hard to keep moving back and forth between the text and the figures.

Why does "None" have 2 colours in Fig. 2?

>Note that to surpass a zero score, both agents required kickstarting [29], with agents first trained on a fixed chemistry with shaping rewards included for each potion use.

It was not made clear that "both agents" meant the image-based agents here.

>Informal inspection of trajectories from the symbolic version of the task was consistent with the conclusion that the agent, like the random heuristic policy, was dipping stones in potions essentially at random until a high point value happened to be attained.

When exactly does an episode end is not clear to me?


2) **The paper could do a better job of describing its limitations**

Appendix E mentions "thorough discussion of limitations" but I did not see this.

In my opinion, the tasks are still some very specific tasks and cannot cover the whole range of tasks we see in the real world. This is fine but should be mentioned in the limitations.

The environment is very expensive. It seems to me that there is no real signal even until one billion timesteps in Figure 7. This should also be mentioned in the limitations.


**Additional Feedback:**

I am curious to know how the hyperparameters were set. Given how expensive the environment is, this information is valuable I believe.


**Clarity:**

I had already read the arxiv version when it came out but it still took me a while to properly understand the many things mentioned above. But apart from those, the paper *is* mostly well-written.

**Documentation:**

I browsed the Github repo and it looks good to me.

**Ethics:**

Not applicable.

**Relation To Prior Work:**

I think the comparisons with related work could have been more detailed and possibly a section of their own.

**Summary And Contributions:**

The paper describes a 3D video game environment developed in Unity. The environment has an underlying latent causal structure that needs to be identified at the meta-level for an agent to perform well. The authors argue that their environment is accessible (ground truth available) and interesting (similar to challenging real-world tasks featuring compositionality, causal relationships, and opportunities for conceptual abstraction with solutions requiring strategic sequencing of actions). The environment can be image-based (3D) or feature-based (symbolic). They show that recent strong algorithms fail to identify the latent causal structure and how augmenting image-based and feature-based versions of one of the algorithms with various forms of "privileged information" gets their performances closer to an oracle's.

---

> ### Author Response · Authors · 2021-09-28
> **Thank you for your suggestions (part 1)**
>
> We thank you for taking the time and effort to provide such useful detailed feedback regarding clarity and other issues, and also thank you for the kind words regarding the novelty, interestingness, and presentation. We have updated the paper and hope you find the clarity much improved. Please find below and point-by-point response to all comments raised:
>
> > The clarity of the paper could be improved in many places
> > It is not clear which bar "agent" in the above sentence refers to in Fig. 3a and thus it is difficult to parse the figure. Why not already call the agent by the name used in the legend of the plot? For example, "Input: Ground truth" and "Input: Belief state" haven't been defined in the text yet at this point and I found it hard to make out if "Input: Ground truth" may be the oracle or the ideal observer. This is especially so because the oracle or the ideal observer are not mentioned in the figure caption until the end and I had read the caption only till where it describes sub-figure "a" because the text refers to Fig. 3a. I would recommend to add the ideal observer and oracle also to the graph legend instead of just in the caption.
>
> In the course of trying to make figures concise, we perhaps made them too confusing. We have now split up figure 3 to create a new figure 2 (previous figure 3 is now figure 4) and make it clear that at this point in the text, we’re only talking about the baseline agent, which has no additional augmentations or privileged information. We now also clearly label the ideal observer, random heuristic, and oracle lines in the legend in this and other figures. We also made the terminology consistent between the text, figures, and captions. Thank you for your suggestions!
>
> >I think I know why the oracle is only shown in sub-figure a in Fig. 3 but it would be nice to add the reasoning behind this to the caption because it took me a while to figure it out.
>
> We now explain in the caption that because the oracle doesn’t maintain a belief distribution (as it has no uncertainty) it can’t be added to panels c and d, and because it needs to do no exploration, its line is just the 0 line for panel b.
>
> > I did not understand the "Synthetic models" paragraph. Especially, what "fit of these two models" and "was compared for the behavior of the ideal observer reference agent".
>
> Thank you for pointing this point of confusion out. We can see how use of the word “model” can be ambiguous here. We’ve replaced that term with “synthetic probabilistic policies,” as these are hand-crafted probabilistic policies that aren’t models in the same sense that deep RL agents are (although are referred to as models in the cognitive science literature).
>
> > Additionally, the sentence
> >> “In both models, predictions were based on a posterior distribution over the current chemistry.”
>
> > could just say that the input was the belief state. Then the terminology would be consistent with the figure caption. in my opinion, scientific papers should try to use consistent terminology to reduce the cognitive load of understanding them.
>
> These cognitive models are actually not trained through deep RL. They update their policy based on observations, and are adaptive, but have very hand-crafted update mechanisms. Because they are hand-crafted, we can imbue them with different types of knowledge. By comparing which one our agents are more similar to, in terms of maximum likelihood, we can make inferences about the kinds of knowledge that our agents might possess.
>
> We’ve now added clarifying language in this section and have made the terminology consistent. Hopefully this is now clearer in the text, but please let us know if it’s not.
>
> > Did you mean Fig. 2b? I don't see this in 2a (this error is repeated multiple times later). Again, it could have been mentioned here that the reader should look at the "ground truth" bar in the left half of 2b instead of saying "side information". The "ground truth" aspect is only mentioned in the next sentence but I was already trying to parse the figure after reading the quoted sentence above.
>
> Thank you for spotting this! We reorganized the figures to improve clarity at one point and didn’t spot all the changed references. We also made the terminology consistent between the text, figures, and captions.
>
> > Compositionality was not discussed in the text. I would like that it is described why Alchemy tasks feature compositionality.
>
> We consider Alchemy to be compositional because different components of the chemistry (as well as other task aspects) are sampled independently in the procedural generation process (see section A.1 in the appendix). We emphasized this more throughout the text (task description and related work sections).

---

> > ### Author Response · Authors · 2021-09-28
> > **Thank you for your suggestions (part 2)**
> >
> > > I also find the term "3D Alchemy" misleading because I think the authors mean the image-based task here but according to me even the symbolic version is inherently describing a 3D world and thus I find "3D Alchemy" a misnomer.
> >
> > Although symbolic Alchemy describes the same underlying causal mechanisms and MDP as the 3D version, because the action space is much simpler (actions taken immediately apply a potion to a stone, or deposits the stone into the cauldron), and all stones and potions are visible at once, we don’t really consider it to be 3D, since this would imply taking in observations and actions in a 3D space. We’ve added clarifying language to the symbolic Alchemy task description.
> >
> > > Why does it require longer sequences?
> >
> > In 3D, the agent has to navigate around a simulated space and manipulate stones with an actuator. This means that there are many more steps between potion applications and rewards. However, in symbolic Alchemy, potion applications are single-step actions (see previous answer for more detail).
> >
> > > What does "individual replicas" mean? Seeds? Why are they 5-8 and not a fixed number?
> >
> > Yes, individual replicas are random seeds. The experiments were run at different times with different amounts of resources available. We chose to report all replicas rather than artificially limit our reporting to 5 for each condition.
> >
> > > In line 300, please mention what part of the Appendix.
> >
> > > Sentence beginning in lines 300-306 could be improved grammatically.
> >
> > > Citation needed.
> >
> > Thank you, we have made all of these changes.
> >
> > > Figures 3 and 4 are discussed before Figure 2. And it is hard to keep moving back and forth between the text and the figures.
> >
> > We’ve now added a new figure 2 which should ameliorate this issue. All figures are now mentioned in order.
> >
> > > Why does "None" have 2 colours in Fig. 2?
> >
> > This was because we previously wanted to highlight that the light blue agents are the “baseline” agents, without any enhancement. However, we understand that this is confusing, so have now removed it. With the addition of the new figure 2, we believe the point to be much clearer, thank you for your suggestions!
> >
> > > It was not made clear that "both agents" meant the image-based agents here.
> >
> > This has now been clarified.
> >
> > > When exactly does an episode end is not clear to me?
> >
> > Episodes end at the end of 10 trials, with a fixed number of steps per trial. There are 20 steps per trial for symbolic Alchemy, and 18000 frames (at a frame rate of 30 fps, and thus 1 minute trials) for full Alchemy. We’ve added this information to the appendix, section A.2, thank you for pointing out the oversight.
> >
> > > Appendix E mentions "thorough discussion of limitations" but I did not see this.
> >
> > > In my opinion, the tasks are still some very specific tasks and cannot cover the whole range of tasks we see in the real world. This is fine but should be mentioned in the limitations.
> >
> > > The environment is very expensive. It seems to me that there is no real signal even until one billion timesteps in Figure 7. This should also be mentioned in the limitations.
> >
> > Thank you for pointing all of these out. We’ve added a paragraph to the discussion regarding these and other points.
> >
> > > I think the comparisons with related work could have been more detailed and possibly a section of their own.
> >
> > We have now added an entirely new, separate section for related work, also in line with another reviewer’s feedback.
> >
> > > I am curious to know how the hyperparameters were set. Given how expensive the environment is, this information is valuable I believe.
> >
> > We conducted coarse hyperparameter searches  in order to balance resource constraints against the complexity of the task, doing sweeps (2-3 values) over learning rate, memory length of the transformer, and agent-specific hyperparameters such as the MPO epsilon temperature and target update period for VMPO. We have updated the appendix to include this information.
> >
> > We should note that for symbolic alchemy, many agents converged a quarter of the way through training, but were run for much longer in order to ensure that all agents received the same amount of training before evaluation. Further, there are many ways to simplify Alchemy considerably so that it can be learned much faster, such as holding various generative factors constant, or reducing the number of stones / increasing the number of trials. These are all possible given that the environment and analysis tools have been released publicly. This is now discussed in more detail in the paper.
> >
> > Finally, we should note that the agents we tested, VMPO and Impala, are general purpose deep RL agents designed for completely different tasks, and so we wouldn't expect them to be extremely efficient at Alchemy. New agents designed to solve the meta-learning challenge proposed by Alchemy should be able to be much more data efficient.

---

> > > ### Comment · Reviewer_WAyX · 2021-10-04
> > > **Increasing score**
> > >
> > > Thank you very much to the authors for their response and improvements made to the paper! I increase my score by 1.

---

> > > > ### Comment · Reviewer_WAyX · 2021-10-04
> > > > **Edit button for main review seems disabled**
> > > >
> > > > Clarification: I increase my score from 7 to 8. But, the edit button for my main review seems disabled on OpenReview, so it still shows as a 7.

---

> > > > ### Author Response · Authors · 2021-10-04
> > > > **Thank you for your re-evaluation!**
> > > >
> > > > We thank the reviewer for kindly taking the time to re-evaluate our paper and updating their rating, and thank them again for their very useful suggestions!

---

### Official Review · Reviewer_1Tpo · 2021-09-21
**Clarifications are needed**

**Rating:** 5
**Confidence:** 4
**Clarity:** The paper is easy to follow.

**Strengths:**

The structure information is interesting for RL methods. The implementation of Alchemy gives the features via the generating graph topology. The experiments on VMPO and IMPALA demonstrate that the tasks are challenging for the two algorithms. The author also gives ablation studies.

**Weaknesses:**

1. As far as I understand, this paper proposes 1 environment. So how representative the environment is should be clarified, e.g., does it contains properties that many other tasks may share.

2. The task generating method, i.e., the task is resampled at the start of each episode, is only one possible setting for meta-RL. However, such a setting is unfriendly for RL training, since it creates a dynamic environment that can injure the learning. Another setting where the sampled tasks are fixed at the beginning is thus a common choice.

3. As a benchmark, this paper should survey more about related benchmarks, and test more meta-RL methods.

4. The paper show that the environment requires a good latent structure to be learned in order to achieve a good performance. The environment is interesting for validating latent structure learning. So, why not focusing on proposing an environment for structure RL? One reading the paper is hard to feel this paper is related to meta-RL, which addressing questions like what to extract from past tasks and how to transfer to new tasks. These questions seems orthogonal with learning latent structure.

**Additional Feedback:**

none

**Correctness:**

It seems that there is no hyper-parameter search. Not sure if the empirical conclusions still hold after a well search.

**Documentation:**

have given document in the github link.

**Ethics:**

not found anything violating ethics yet.

**Relation To Prior Work:**

The prior work of environments designed for meta-learning should be discussed in detail.

**Summary And Contributions:**

The paper introduces a new benchmark for meta-RL research, emphasizing transparency and potential for in-depth analysis and structural richness. The proposed Alchemy environments give the two features: 1) structural interestingness, which features compositionality, causal relationships, and opportunities for conceptual abstraction; 2) structural accessibility, in which the ground-truth parameterization of the distribution should ideally be accessible. The authors also conduct experiments with two meta-learning algorithms in the Alchemy environments. The results show that the agents failed in structure learning and latent-state inference and thus displayed poor performances in the Alchemy environments.

---

> ### Author Response · Authors · 2021-09-24
> **Points of clarification (1/2)**
>
> Thanks to the reviewer for taking the time and effort to consider our paper. We hope the following points of clarification help in making further assessment.
>
> > As far as I understand, this paper proposes 1 environment. So how representative the environment is should be clarified, e.g., does it contains properties that many other tasks may share.
>
> Alchemy was designed with inspiration from cognitive science studies of human behavior, especially work by Griffiths, Tenenbaum et al. (such as Tenenbaum 2011 Science), where it is argued that latent structure learning is generic to real-world tasks/intelligence (with structure very much like what we built into Alchemy). We therefore believe it to share many properties with tasks that evoke human-like cognition, such when children learn to experiment in their environment to learn about causal dependencies (Gopnik, 2004 Psych Review), or tasks that require theory formation, causal understanding, or planning in a latent space (Dasgupta et al, 2019 arXiv:1901.08162; Tsividis et al, 2021 arXiv:2107.12544). These kinds of properties are notably absent in the most popular meta-RL benchmarks today, which mostly focus on generalization of simulated robotic control policies (ie Mujoco continuous control tasks or MetaWorld).
>
> We also note that, while Alchemy is one environment, due to its procedurally generated nature, there are actually many possible versions that can be played in a given episode, unlike, for instance, Atari. Training on procedurally generated tasks has been shown to improve generalization performance and reduce overfitting (Risi and Togelius, 2020 Nat Mach Int; Cobbe et al, 2019 ICML).
>
> We will add all of these points to a new, separate “related works” section to be included in the camera-ready (if accepted) and thank the reviewer for bringing up this important point of confusion, which wasn’t made clear previously.
>
>
> > The task generating method, i.e., the task is resampled at the start of each episode, is only one possible setting for meta-RL. However, such a setting is unfriendly for RL training, since it creates a dynamic environment that can injure the learning. Another setting where the sampled tasks are fixed at the beginning is thus a common choice.
>
> Is the reviewer referring to the issue of catastrophic forgetting when training on only a few tasks one at a time? This issue doesn’t apply since we have many more possible configurations, and are resampling every episode.
>
> We note that according to the classic definition of learning to learn (Thrun and Pratt, 2012), training on a distribution (or family) of tasks with shared structure (as is present in Alchemy) allows for improved performance and adaptation with increased experience and number of tasks. There is a strong literature investigating under which conditions models can learn to adapt to related tasks, in a meta-learning sense (Schmidhuber, 1987; Thrun & Pratt, 2012; Naik & Mammone, 1992; Bengio et al., 1991; 1992; Hochreiter et al., 2001; Andrychowicz et al., 2016; Santoro et al., 2016).
>
> Training in an environment where all tasks are initialized the same way every episode, on the other hand, would lead to overfitting and lack of generalization. Therefore Alchemy is procedurally generated every episode to have some randomness (see above). At the same time, crucially, there are some properties which are kept fixed throughout an episode (such as the chemistry), and certain generative properties that are invariant throughout training (see section 2.2 Chemistry). These invariances are what are meta-learned and allow for faster adaptation throughout training.
>
> That said, Alchemy does come with “easier” levels in which the chemistry can be fixed throughout training, which we used in part to “kickstart” our agents to be able to obtain nonzero reward in the full, 3D version.

---

> ### Author Response · Authors · 2021-09-24
> **Points of clarification (2/2)**
>
> > As a benchmark, this paper should survey more about related benchmarks, and test more meta-RL methods.
>
> Due to space constraints, we integrated related work throughout the introduction. If accepted, we will make use of the extra content page to add a separate, more fully fleshed out related works section.
>
> With regard to testing more meta-RL methods, we note that the main purpose of our work is to provide a tool for researchers which can diagnose their agents’ ability to perform latent state inference. To this end, we provide several comparison strategies such as the oracle, heuristic strategies, and a Bayesian ideal observer. Comparing agents of interest with these bot-like, non-learning strategies, which have been handcrafted with specific knowledge and abilities, is actually more informative than comparison against other complex, deep RL algorithms, which require hyperparameter / architecture tuning and varying amounts of compute. Our main purpose of benchmarking Impala and VMPO was to gauge whether otherwise highly performant agents (which have been shown to excel at other tasks) could solve Alchemy, as well as to showcase the analysis capabilities included. This is not to say that current, existing approaches could not solve Alchemy; in fact we would welcome it if someone could show this, as we believe that this would be highly informative and could shed some light on exactly which capabilities are missing from the agents we’ve tested, and how to enhance them. We’ve already been in contact with external researchers who have applied Alchemy to various agents, indicating its feasibility and usefulness to the wider community.
>
> Note that due to the fact that Alchemy requires extensive within-episode adaptation, having some form of recurrent memory is vital, and hence approaches like DQN and MAML simply would not work.
>
> > The paper show that the environment requires a good latent structure to be learned in order to achieve a good performance. The environment is interesting for validating latent structure learning. So, why not focusing on proposing an environment for structure RL? One reading the paper is hard to feel this paper is related to meta-RL, which addressing questions like what to extract from past tasks and how to transfer to new tasks. These questions seems orthogonal with learning latent structure.
>
> In our understanding, meta-learning and latent structure learning are in fact intimately related, as it is the structure in the tasks that allows for fast learning and adaptation in the inner loop, arising through meta-learning. See for example Grant et al, 2018 (arxiv:1801.08930), which demonstrated that gradient-based meta-learning is essentially performing hierarchical Bayesian modeling (a form of probabilistic inference) and thus is learning latent structure. Mikulik et al 2020 (NeurIPS) and Ortega et al 2019 (arXiv:1905.03030) have shown that black-box memory-based meta-learners (implemented using recurrent memory of some form rather than gradient-based optimization) are in the limit of training able to implement Bayes-optimal policies due to their meta-learning objective. Analysis of their internal representations shows that they come to represent the task in a similar way to Bayes-optimal agents, capturing the necessary and latent structure of the task family.
>
> If the reviewer has a different perspective or if we’ve misconstrued the question, we’d very much be open to discussing.
>
>
> Hyperparameter selection:
>
> We conducted coarse hyperparameter searches  in order to balance resource constraints against the complexity of the task, doing sweeps (2-3 values) over learning rate, memory length of the transformer, and agent-specific hyperparameters such as the MPO epsilon temperature and target update period for VMPO. We will update the appendix to include this information.
>
> Most of the hyperparameters were taken either from the original VMPO and Impala papers, or swept in consultation with the authors, based on their best estimate of what could work for Alchemy.
>
>
> We hope that this cleared some things up for you and look forward to your response!

---

> ### Author Response · Authors · 2021-10-03
> **Gentle reminder**
>
> We would like to provide a gentle reminder to the reviewer that we have uploaded a revised version of our paper, which we believe completely addresses the concerns expressed in their review, in the following ways:
>
> 1. Adds a new related works section which significantly fleshes out our survey of current popular meta-RL benchmarks (adding over 30 references)
> 2. Highlights the procedurally generated nature of Alchemy, as well as contextualizes the use of procedural generation in meta-RL
> 3. Provides background on the relation between latent structure learning and meta-learning
> 4. Foregrounds the way in which Alchemy is inspired by cognitive science-style tasks and analyses, and how it shares properties with these kinds of tasks
> 5. Adds details regarding the hyperparameter search process to the appendix.
>
> We are happy to answer any follow up questions or concerns!
>
> Thank you,
> Authors

---

### Author Response · Authors · 2021-09-29
**Revision uploaded**

A major thank you to all the reviewers for their time and thoughtful feedback.

We’ve uploaded a revised version of the paper, which we hope reviewers will agree is much improved.

The following is a summary of the major changes made (see a detailed list of changes in individual response to reviewers):

1. A new related works section has been added, per reviewer 1Tpo and WAyX’s suggestion, which dramatically fleshes out discussion of existing meta-RL benchmarks, as well as other background regarding meta-learning and Alchemy’s cognitive science motivations, in response to reviewer WeQ5.
2. We’ve added a paragraph on limitations in the discussion, per reviewer’s WAyX’s suggestion.
3. A new figure 2 has been created, in order to distinguish baseline results, constituting the Alchemy benchmark itself, from the augmentation results. This also created better organization among the figures, per WAyX’s suggestion.
4. Many fixes to the figures and text were added to improve clarity, in response to points of confusion raised by all reviewers.
5. We’ve added information about hyperparameter selection and other details about episode structure to the appendix.

If there are any additional points we can address, please let us know. Thank you for your consideration.

---

### Decision · Program_Chairs · 2021-10-09

**Decision:**

Accept

**Comment:**

This work introduced a Unity-based 3D environment for evaluating the ability of meta-reinforcement learning agents to infer latent causal structures from interactions. At the initial reviews, this work received 5, 7, 7 from three expert reviewers. Reviewer 1Tpo requested clarifications on task design and survey about related benchmarks. Reviewer WAyX and WeQ5 felt positive but also raised several clarification questions about the technical details. The authors provided detailed responses to these questions and addressed the reviewers' comments well. In the end, Reviewer WeQ5 agreed to upgrade their rating from 7 to 8 but didn't get a chance to do so before the review system is closed in the final phase. The AC read the paper, the reviews, and the authors' discussions in detail, and agreed with the reviewers that this paper introduced an interesting new benchmark to facilitate research in the challenging meta-RL domain. Taking all the factors into account, we recommend accepting this paper in this track.